# Angiogenic CD34 Stem Cell Therapy in Coronary Microvascular Repair—A Systematic Review

**DOI:** 10.3390/cells10051137

**Published:** 2021-05-08

**Authors:** Balaj Rai, Janki Shukla, Timothy D. Henry, Odayme Quesada

**Affiliations:** 1Lindner Center for Research, The Christ Hospital, Cincinnati, OH 45219, USA; Balaj.Rai@thechristhospital.com (B.R.); tim.henry@thechristhospital.com (T.D.H.); 2Department of Internal Medicine, University of Cincinnati Medical School, Cincinnati, OH 45219, USA; shuklajc@mail.uc.edu; 3Women’s Heart Center, Vascular and Lung Institute, The Christ Hospital, Cincinnati, OH 45219, USA

**Keywords:** CD34 stem cell therapy, coronary microvascular dysfunction, refractory angina, ischemia with non-obstructive coronary arteries

## Abstract

Ischemia with non-obstructive coronary arteries (INOCA) is an increasingly recognized disease, with a prevalence of 3 to 4 million individuals, and is associated with a higher risk of morbidity, mortality, and a worse quality of life. Persistent angina in many patients with INOCA is due to coronary microvascular dysfunction (CMD), which can be difficult to diagnose and treat. A coronary flow reserve <2.5 is used to diagnose endothelial-independent CMD. Antianginal treatments are often ineffective in endothelial-independent CMD and thus novel treatment modalities are currently being studied for safety and efficacy. CD34^+^ cell therapy is a promising treatment option for these patients, as it has been shown to promote vascular repair and enhance angiogenesis in the microvasculature. The resulting restoration of the microcirculation improves myocardial tissue perfusion, resulting in the recovery of coronary microvascular function, as evidenced by an improvement in coronary flow reserve. A pilot study in INOCA patients with endothelial-independent CMD and persistent angina, treated with autologous intracoronary CD34^+^ stem cells, demonstrated a significant improvement in coronary flow reserve, angina frequency, Canadian Cardiovascular Society class, and quality of life (ESCaPE-CMD, NCT03508609). This work is being further evaluated in the ongoing FREEDOM (NCT04614467) placebo-controlled trial.

## 1. Introduction

Ischemia with non-obstructive coronary arteries (INOCA) is an increasingly recognized disease with an estimated prevalence of 3 to 4 million individuals [1]. INOCA is characterized by signs and symptoms of ischemia in the absence of obstructive disease [2,3,4,5,6,7]. Although the pathophysiology is not completely understood, coronary microvascular dysfunction (CMD) has been shown to play a critical role and is reported in 47–64% of INOCA patients [1,5,8,9,10,11]. CMD encompasses endothelial-dependent and endothelial-independent microvascular dysfunction. Endothelium-dependent CMD stems from the inability of endothelial cells to produce vasodilatory substrates, thus blunting adequate myocardial perfusion during stress. Conversely, endothelial-independent CMD is detected as abnormal microvascular dilation resulting in a reduced coronary flow reserve (CFR) in response to adenosine. Endothelium-independent CMD results from the inability of smooth muscles to dilate despite the presence of vasodilatory substrates. CMD is associated with increased rates of major adverse cardiac events (MACE), including heart failure (HF), myocardial infarction (MI), and non-fatal stroke during long-term follow-up [2,3,4,11]. However, the current treatment is limited to risk factor management and antianginal medications, with no specific therapy for ischemia, which can significantly limit quality of life.

Cell therapy using autologous stem cells expressing CD34 (CD34^+^) is a novel therapeutic option for INOCA patients with CMD and refractory angina given the ability of CD34^+^ cells to repair the microcirculation [12,13,14]. Data from animal models indicate that CD34^+^ stem cells differentiate into endothelial cells, which incorporate into new vasculature and aid in the release of angiogenic cytokines, thus promoting vascular repair in the microcirculation, leading to improved myocardial perfusion in tissues damaged by acute and chronic ischemia [12,13,14,15,16,17,18]. Three consecutive, randomized, double-blinded, placebo-controlled trials in the United States, in patients with obstructive coronary artery disease (CAD) and Canadian Cardiovascular Society (CCS) class 3–4 refractory angina, established the feasibility of the intra-myocardial delivery of auto-CD34^+^ cells, showing significant improvements in short- and long-term anginal frequency (AF) and total exercise time (TET) [15,16,18]. Meta-analysis of these three trials demonstrated significant improvement in AF and TET, and a reduction in mortality [19]. PreSERVE-AMI, a large clinical trial using CD34^+^ cell therapy for left ventricular dysfunction post-ST-segment elevation MI (STEMI), demonstrated the safety and potential efficacy of the intracoronary infusion of CD34^+^ cell therapy in post-MI patients [20]. A recent two-center, phase 1 feasibility and safety trial (ESCaPE-CMD, NCT03508609) using autologous CD34^+^ stem cells in INOCA patients with endothelial-independent CMD demonstrated a significant improvement in CFR, AF, CCS class and quality of life [21]. These results led to an ongoing, double-blinded, placebo-controlled trial (FREEDOM, NCT04614467) to demonstrate the safety and efficacy of the intracoronary delivery of CD34^+^ stem cells. We will review the preclinical and clinical data that led to the development and application of cell-based therapy for ischemic repair, with a focus on microvascular repair in INOCA patients with CMD.

## 2. Discovery of CD34^+^ Cells as Endothelial Progenitor

CD34^+^ is a transmembrane phosphoglycoprotein and a cell surface marker common to many hematopoietic progenitor cells. Prior to their use in ischemic disease, CD34^+^ stem cells were used in hematopoietic reconstitution after myeloablative therapy in many cancer patients [22]. In 1997, a landmark in vitro study showed that CD34^+^ stem cells differentiate into cells, which express endothelial cell characteristics [14]. In this study, CD34^+^ mononuclear blood cells were isolated from human peripheral blood and plated on fibronectin, resulting in spindle shaped attachments. When grown in co-culture with CD34-negative (CD34^−^) cells, they formed clusters resembling epiblast blood islands, which gave rise to endothelial cells and vascular structures in vitro, and expressed endothelial lineage markers such as CD34, CD31, Flk-1, Tie-2, and E-selectin. This suggested that CD34^+^ stem cells could play a role in adult angiogenesis and have wider applications than solely hematopoiesis. There are two main theoretical mechanisms of angiogenesis by CD34^+^ stem cells: [12,13] (1) direct incorporation of CD34^+^ cells into new vasculature [14]; (2) release of angiogenic cytokines in a paracrine manner [23].

## 3. Preclinical Studies on CD34 Therapy for Ischemic Disease

Following bench-top studies, human CD34^+^ cells were administered into athymic mouse and rabbit models of hindlimb ischemia. Athymic models were used to avoid a host immune response and allow the use of human cells rather than rodent cells. Histologic examination showed that labeled human CD34^+^ cells appeared integrated into 13.4 ± 5.7% of the capillary vessel walls of athymic hindlimb ischemic mice, compared to 1.6 ± 0.8% of labeled human CD34^−^ cells. These cells localized only to the ischemic limb [12,24]. Through these studies, researchers showed that CD34^+^ stem cells target ischemic tissue and incorporate physically into new vessel walls.

The next step in the use of CD34^+^ stem cells came from a study of CD34^+^ cells in diabetic mice [25]. Labeled human CD34^+^ stem cells were injected into mice following ligation and transection of the femoral artery. Diabetic mice with CD34^+^ injections had a significantly higher rate of flow restoration as compared to mice injected with CD34^-^ cells, not seen in nondiabetic mice. Additional experimentation to determine a causal link was conducted, but no link with insulin or glucose was established; therefore, why exogenous CD34^+^ cells helped accelerate flow restoration in diabetic, but not nondiabetic, mice remains unclear. Over the following years, several studies established that CD34^+^ cells significantly augmented perfusion in hindlimb ischemia models [26,27]. 

These findings were extended from peripheral ischemia to myocardial ischemia [28]. The injection of human CD34^+^ stem cells in mouse infarction resulted in a significant increase in infarct zone microvascularity and cellularity, and a significant decrease in matrix deposition and fibrosis. Additionally, this study found that the left ventricular ejection fraction (LVEF) of ischemic mice treated with CD34^+^ stem cells recovered by a mean of 22% (*p* < 0.001) [28]. These studies established that not only did CD34^+^ cells increase angiogenesis, but they also led to functional improvement in the myocardium.

This work was continued in four cynomolgus monkeys receiving autologous CD34^+^ stem cells transplanted to the pre-ischemic zone after ligation of the left anterior descending artery (LAD) [23]. Compared to the control group, the transplant group had improved regional blood flow and cardiac function. However, this study reported that only a few of the new endothelial cells were progeny of the transplanted cells. Additionally, the study showed that the population that received CD34^+^ stem cells had significantly increased regional vascular endothelial growth factor levels [23]. This raised the possibility of a second mechanism of action of CD34^+^ stem cells–angiogenic cytokine recruitment via paracrine signaling to induce angiogenesis, rather than direct incorporation. This mechanism is supported by earlier in vitro cell studies showing that CD34^+^ cells express multiple growth factors, cytokines, and chemokines associated with hematopoiesis and angiogenesis [29,30]. 

## 4. Clinical Studies on CD34 Therapy for Ischemic Disease

Although the majority of the clinical trials have been in patients with no-option refractory angina [15,16,17,18,19], CD34^+^ stem cells have also been investigated in clinical studies for peripheral ischemia [31,32], nonischemic cardiomyopathy [33], myocardial infarction [20], and ischemic stroke [34,35].

### 4.1. CD34 Therapy for Peripheral Ischemia

In peripheral ischemia, phase 1 and phase 2 trials have shown statistically significant improvements in walking distance, limb pain, blood flow, tissue oxygenation, reduction in ulcer size, and amputation-free survival in no-option critical limb ischemia [12,13,31]. A small phase 2 trial enrolled six patients with Rutherford category 4 or 5 critical limb ischemia. All but one of the patients showed an improvement in Rutherford category from baseline that was statistically significant at 24 weeks and 52 weeks. Currently, a phase 3 trial using CD34^+^ stem cells for critical limb ischemia due to atherosclerosis and Buerger’s disease is near completion in Japan [32].

### 4.2. CD34 Therapy for Nonischemic Cardiomyopathy

In nonischemic dilated cardiomyopathy, an open-label randomized study enrolled 110 patients, including 55 treated with intracoronary delivery of CD34^+^ stem cells [33]. The five-year follow-up showed that CD34^+^ patients underwent significant improvement in cardiac function and exercise capacity, reduction in N-terminal-pro B-type natriuretic peptide, and an excellent safety profile. The five-year survival by Kaplan–Meier analysis was 2.3 times higher in the treatment group than the control group (*p* = 0.015). Additionally, 20% of the cell solution was labeled with a radioisotope tracer to determine intramyocardial homing. Interestingly, patients with poor homing did not show improvements in LVEF, likely due to a downregulation of homing factors in dilated cardiomyopathy, leading to poor stem cell retention in the myocardium.

### 4.3. CD34 Therapy for Myocardial Infarction

PreSERVE-AMI was a randomized, double-blinded, placebo-controlled, phase 2 study using intracoronary administration of bone marrow-derived autologous CD34^+^ cells in patients with left ventricular dysfunction post-STEMI [20]. The study enrolled 161 patients randomized to CD34^+^ cells versus placebo, with a primary efficacy endpoint of improvement in the mean resting total severity score by single-photon emission computerized tomography (SPECT) scan. Although the mean resting total severity score improved in both groups, the mortality rate was significantly lower in the CD34^+^-treated patients. There was no significant difference in overall improvement in LVEF or infarct size between the treatment and control groups. However, there was a significant relationship between CD34^+^ cell dose and the change in infarct size, LVEF, and days alive, with a benefit in those with higher dosages when adjusted for ischemic time. This study suggested that CD34^+^ stem cell therapy may improve outcomes in selected sub-populations of STEMI patients, even though it did not meet the primary efficacy endpoint. 

### 4.4. CD34 Therapy for Ischemic Stroke

A phase 1 trial demonstrated the safety and feasibility of bone marrow-derived CD34^+^ cells delivered intra-arterially to stroke patients [34]. This nonrandomized, open-label, prospective trial enrolled five patients presenting within 7 days of onset of severe anterior circulation ischemic stroke. All patients had clinical improvement as assessed by modified Rankin score (median score: 4 to 2) and National Institutes of Health Stroke Scale (median score: 9 to 2) at 6 months with no significant adverse events, thus establishing preliminary safety and feasibility. Furthermore, the ongoing STROKE34 trial is a randomized, placebo-controlled, phase 2a trial using CD34^+^ stem cells in patients with acute ischemic stroke due to occlusion of the middle cerebral artery [35]. 

### 4.5. CD34 Therapy for Refractory Angina

Successful phase 1 and phase 2 trials have shown the safety and efficacy of intramyocardial CD34^+^ stem cell therapy in treating no-option refractory angina. The phase 1, double-blinded, placebo-controlled trial enrolled 24 patients with CCS class 3 or 4 angina on optimal medical treatment, and demonstrated initial safety as well as clinical improvement in the CD34^+^ treated patients [15]. Under physiological conditions, the circulating CD34^+^ cell concentration is too low in the peripheral circulation. Thus, stimulation with granulocyte colony-stimulating factor (G-CSF) at 5 μg/kg per day for 4–5 days, followed by leukapheresis at day 5 with subsequent CD34^+^ enrichment, was used to harvest autologous mobilized CD34^+^ cells. The phase 2 ACT 34 trial enrolled 167 patients who were randomized to receive CD34^+^ stem cells in low doses (1 × 10^5^ cells/kg, n = 55), high doses (5 × 10^5^ cells/kg, n = 56), and an equal volume placebo diluent (n = 56) [16]. Treatment was distributed between 10 distinct ischemic sites with viable myocardium using a NOGA Myostar^®^ mapping injection catheter. Both CD34^+^ cell therapy-treated patient groups showed significant improvements in the primary endpoint of AF at 6 months and 12 months. Compared to the placebo group, the low-dose treatment group showed a reduction in AF (6 months: 6.81 vs. 10.91 episodes per week, *p* = 0.02; 12 months: 6.3 vs. 11.0 episodes per week, *p* = 0.035) and improvement in TET at 6 and 12 months (6 months: 139 ± 151 vs. 69 ± 122 s, *p* = 0.014; 12 months: 140 ± 171 vs. 58 ± 146 s, *p* = 0.017) [16]. A two-year follow-up of 130 of those patients showed that autologous CD34^+^ cell therapy was associated with persistent improvements in AF in both the low-dose and high-dose groups (*p* = 0.03) [17]. Additionally, there was a decrease in mortality (*p* = 0.08) and MACE (*p* = 0.08). 

The phase 3 RENEW, randomized, double-blinded, placebo-controlled trial was originally designed to obtain FDA approval using improvements in TET as the primary endpoint. Unfortunately, the trial was stopped early due to financial issues with the sponsor. Among the 112 enrolled patients, the improvement in TET was 61.0 s at 3 months (95% confidence interval (CI): −2.9 to 124.8, *p* = 0.06), 46.2 s at 6 months (95% CI: −28.0 to 120.4, *p* = 0.22), and 36.6 s at 12 months (95% CI: –56.1 to 129.2, *p* = 0.43) [18]. Furthermore, AF was significantly improved at 6 months (relative risk (RR): 0.58 by intention to treat, *p* = 0.02). As the study was incomplete, researchers were unable to conclusively determine the efficacy of CD34^+^ cell therapy for refractory angina patients. Further retrospective data analyses have shown that CD34^+^ treatment decreased long-term mortality (24% vs. 47%, *p* = 0.02), costs related to cardiac care (62% reduction translating to an average of USD 5500, *p* = 0.03), and interventional coronary procedures at 12 months (1.2  ±  0.91 vs. 0.32  ±  0.75 events, *p* < 0.0001) [36]. In all three trials, placebo patients also underwent treatment with G-CSF, leukapheresis, and intramyocardial injections. Therefore, the trials were completely blinded.

A 2018 meta-analysis of these three randomized, double-blinded trials (n = 304)—phase 1 and 2 ACT-34, ACT-34 extension and phase 3 RENEW—showed an improvement in TET, AF, and MACE in patients with obstructive coronary artery disease and refractory angina receiving intramyocardial autologous CD34^+^ cell therapy [19]. TET improved by 46.6 s at 3 months (*p* = 0.007), 49.5 s at 6 months (*p* = 0.016), and 44.7 s at 12 months (*p* = 0.065). Additionally, the relative AF decreased–0.78 at 3 months (*p* = 0.032), –0.66 at 6 months (*p* = 0.012), and –0.58 at 12 months (*p* = 0.011). Lastly, there was a significant decrease in mortality (12.1% vs. 2.5%, *p* = 0.0025) and MACE (38.9% vs. 30.0%, *p* = 0.14) in patients receiving intramyocardial autologous CD34^+^ cell therapy compared to placebo at 24 months (Figure 1).

Although the three previous refractory angina trials used intramyocardial injection, the intracoronary delivery of CD34^+^ stem cells has also been shown to be safe and effective. This potentially allows for a safer and more established delivery route without losing the effectiveness of therapy. A 2010 study from China of 112 patients showed the safety and feasibility of intracoronary CD34^+^ stem cell therapy in refractory angina patients [37]. In this single-center study, there were no differences in the frequency of adverse events between the placebo and treatment groups, showing that the intracoronary delivery of bone-marrow-derived CD34^+^ cells is safely tolerated. Additionally, the study showed that patients receiving intracoronary CD34^+^ stem cell therapy had a greater reduction in AF at 3 months (–14.6 ± 4.8 vs. –4.5 ± 0.3 episodes, *p* < 0.01) and 6 months (–15.6 ± 4.0 vs. –3.0 ± 1.2 events, *p* < 0.01) compared to placebo. Once again, the placebo group also showed a significant decrease in AF from baseline, indicating a strong placebo effect. This study also found a statistically significant decrease in use of nitroglycerin and CCS class, as well as an increase in TET in patients receiving intracoronary CD34^+^ stem cells [37]. In summary, extensive clinical trial data have demonstrated the outstanding safety of CD34^+^ stem cell therapy, and consistently showed clinical improvements in conditions characterized by perfusion abnormalities. No studies to date have reported an increased risk of uncontrolled cell growth of CD34^+^ inoculated tissues, angiomas or cancer.

## 5. Ischemia with Nonobstructive Coronary Artery (INOCA) Disease and Coronary Microvascular Dysfunction (CMD)

Approximately one-half of patients undergoing coronary angiography for known or suspected ischemia are found to have no obstructive coronary artery stenosis, referred to as INOCA. In INOCA, obstructive coronary artery stenosis is commonly defined as >50% narrowing in any major epicardial artery [4]. These patients present with symptoms similar to obstructive CAD, such as angina and angina equivalents including dyspnea, fatigue, and weakness. INOCA is more commonly found in women as compared to men. 

CMD has been found to be the predominant pathophysiologic mechanism in women with INOCA. It includes a spectrum of disorders that affect the coronary microvascular circulation, characterized by endothelial-independent CMD and endothelial-dependent CMD, which limit myocardial perfusion. Endothelium-independent CMD results from a lack of response to adequate endothelial cell-derived vasodilatory substrates, whereas endothelial-dependent CMD stems from the inadequate availability of these substrates. 

The mechanisms contributing to CMD in INOCA are not completely understood, though they appear to be multifactorial [1]. CMD is associated with reduced microcirculatory conductance, caused by microvascular remodeling or dynamic arteriolar obstruction due to vasomotor disorders of the coronary arterioles [3,7]. Pre-arterioles and arterioles regulate a majority of coronary resistance and determine myocardial flow distribution based on oxygen demand. In patients with INOCA and CMD, the narrowing of intramural arterioles and capillaries can lead to microvascular dysfunction. Remodeling instigated by risk factors such as atherosclerosis results in increased wall to lumen ratios and capillary rarefaction, which causes a reduction in the vasodilatory range of the microcirculation, and a subsequent decrease in oxygen delivery to the myocytes. Additionally, remodeled arterioles are hypersensitive to vasoconstrictive stimuli. Under physiological conditions, the dilation of smaller arterioles leads to the upstream vasodilation of larger arterioles and epicardial vessels, which increases oxygen delivery to cardiac myocytes. In CMD, impaired vasodilation or paradoxical vasoconstriction, exacerbated by vasoconstrictor hypersensitivity, can result in worsening anginal symptoms [38]. Furthermore, coronary blood flow is regulated through the modulation of vascular smooth muscles and is affected by various substrates such as serotonin, vasopressin, endothelin, thromboxane, acetylcholine, adenosine, nitric oxide, and norepinephrine [1]. Genetic variants in voltage-gated potassium channels, such as the SUR2 subunit, can also alter vascular tone via a disconnect between intracellular calcium release and changes in membrane excitability due to intracellular metabolism [39].

There is limited correlation between obstructive CAD and the functional impairment of the microcirculation, reflected by CFR [40]. Approximately 70–80% of patients with symptomatic CMD also demonstrate diffuse nonobstructive atherosclerosis by intravascular ultrasound [41]. The Women’s Ischemia Syndrome Evaluation (WISE) trial demonstrated that the five-year risk of MACE was 16.0% in women with non-obstructive CAD, 7.9% in women with normal coronaries, and 2.4% in the asymptomatic control group (*p* < 0.002) [42]. A subsequent follow-up revealed that endothelial-independent CMD characterized by low CFR was an independent predictor of MACE (hazard ratio (HR) 1.2, 95% CI: 1.05–1.38, *p* = 0.008) [11]. Additional studies have shown that the risk of MACE in women is dependent on reduced CFR rather than non-obstructive CAD (adjusted HR: 1.69, 95% CI 1.04–2.76, *p* = 0.03) [43].

### 5.1. Risk Factors in INOCA and CMD

Well-recognized risk factors for obstructive CAD, such as hypertension, diabetes, dyslipidemia, and atherosclerosis, are also associated with an increased risk of CMD through different mechanisms [5]. Hypertension may induce small artery remodeling/stiffening, which contributes to impaired myocardial perfusion, and a subsequent reduction in microcirculation density and increased coronary arteriolar constriction [44]. Chronic hyperglycemia in diabetes can cause endothelium-dependent and endothelium-independent reductions in coronary vasodilatory capacity, potentially through the hyperglycemia-mediated formation of oxygen-derived free radicals, which inactivate endothelium-derived nitric oxide [45]. Dyslipidemia in CMD can precipitate ectopic fat deposition in cardiac myocytes due to a metabolic shift away from free fatty acid metabolism—women with CMD have a higher triglyceride myocardial content and a lower diastolic circumferential strain rate [46]. Higher high-density lipoprotein and lower triglycerides are related to a higher microvascular flow [47]. 

Additionally, CMD is often associated with diffuse non-obstructive atherosclerosis, likely secondary to increased oxidative stress, reduced nitric oxide availability, and an over-activation of the endothelium [48]. The subsequent monocyte recruitment perpetuates the chronic systemic inflammation, which in turn causes endothelial and vascular smooth muscle dysfunction. There has also been a strong association reported between reduced CFR and impaired left ventricular myocardial relaxation or elevated filling pressures, especially in patients with troponin elevations [49]. Additionally, reperfusion damage after coronary stenting can cause microvascular injury, which increases microvascular permeability and results in endothelium-dependent vasoconstriction and impaired vasodilation due to vasoactive substrate washout [39]. It is important to note that typical risk factors are not always present in patients with CMD.

In addition to classic risk factors, other mechanisms, such as hormones and cardiac autonomic dysfunction, can also play a role. Younger premenopausal women exhibit a higher frequency of symptoms, possibly related to fluctuating levels of estrogen. In perimenopausal women, the loss of estrogen results in autonomic dysfunction and worsening anginal symptoms, with a rapid rise in heart rate during exercise [50]. Furthermore, cardiac autonomic dysfunction can result in a decreased smooth muscle sympathetic response to acetylcholine, which may be related to the reduced bioavailability of nitric oxide and prostacyclin, or increased vasoconstriction due to muscarinic hypersensitivity to smooth muscle stimulation [51]. 

### 5.2. Diagnosis of INOCA and CMD

Since the coronary microcirculation is beyond the resolution of invasive or noninvasive coronary angiography, the interrogation of coronary microvascular function with vasoreactivity testing is necessary to establish the diagnosis of CMD. There are several noninvasive and invasive approaches for the evaluation of coronary vasomotor dysfunction, each with advantages and limitations. Conventional testing, such as stress echocardiography or nuclear scintigraphy, are frequently normal, missing the diagnosis due to balanced ischemia, or displaying regional abnormalities that may not follow typical vascular distributions [52].

Noninvasive techniques for endothelial-independent CMD diagnosis include positron emission tomography (PET), cardiac magnetic resonance (CMR), and the less commonly used modalities such as doppler echocardiography and dynamic myocardial perfusion computerized tomography (CT). These imaging modalities rely on measuring regional and global myocardial blood flow at rest and during stress to calculate CFR [53]. PET is the most accurate and validated imaging modality, and is considered to be the gold standard for non-invasive diagnosis of CMD based on a diagnostic threshold of CFR < 2 [54]. CFR_PET_ <2 is associated with an annual MACE of 7.8% and 5.6% versus 3.3% and 1.7% in men and women, respectively [55]. CMR benefits from a higher spatial resolution, yielding transmural assessments of coronary blood flow as well as a lack of ionizing radiation, and it can be used to detect CMD in patients with INOCA [56]. A myocardial perfusion reserve index <2 has been shown to predict prognosis [57] via the detection of inadequate subendocardial increases in perfusion in response to stress. 

Endothelial-dependent and endothelial-independent CMD are also diagnosed with invasive functional coronary angiography (FCA), previously referred to as coronary reactivity testing (CRT). The European Society of Cardiology Chronic Coronary Syndrome IIa recommend performing guidewire-based measurements of microcirculatory resistance and/or CFR if initial non-invasive testing is unrevealing and symptoms persist [58]. CFR used to diagnose endothelium-independent CMD is the ratio of hyperemic blood flow in response to vasoactive stimuli divided by the resting blood flow. The intracoronary or intravenous administration of adenosine is used to determine CFR via thermodilution or doppler flow velocity. 

The advantage of FCA is that it can be used to diagnose endothelial-dependent microvascular dysfunction in addition to endothelial-independent CMD. Intracoronary acetylcholine (ACH) is used to determine endothelium-dependent CMD and epicardial vasospasm [59]. ACH acts on the muscarinic receptors in endothelial and vascular smooth muscle, resulting in the dilation of normal blood vessels but the paradoxical vasoconstriction of diseased vessels [60]. Arteriolar dysregulation results in reduced vasodilatory response, reduction in blood flow, and the diffuse narrowing of distal epicardial vessels in response to ACH, leading to anginal symptoms. 

Additional measures to assess for CMD include index of microvascular resistance (IMR), hyperemic myocardial velocity resistance (HMR), and fractional flow reserve (FFR). An IMR, calculated as distal coronary pressure at maximal hyperemia multiplied by hyperemic mean index time, greater than or equal to 25 based upon thermodilution is consistent with microvascular dysfunction [61]. An HMR, calculated by dividing intracoronary pressure by hyperemic flow velocity, greater than 1.9 (odds ratio (OR) 15.6, 95% CI: 2.1–114, *p* = 0.007) based upon PET and doppler is an independent predictor of recurrent chest pain [62]. Lastly, flow-limiting CAD can be assessed by FFR, which is the ratio of mean distal coronary pressure to mean aortic pressure at maximal hyperemia; a value of less than or equal to 0.80 is abnormal [63]. Thus, the diagnostic criteria of CMD are FFR > 0.80, CFR < 2.0, IMR ≥ 25, and/or HMR ≥ 1.9 via diagnostic guidewire and adenosine challenge, as well as a < 90% diameter reduction without evidence of angina or ischemic signs on an electrocardiogram (ECG) with vasoreactivity (ACH testing) (Figure 2).

INOCA can also be related to vasospastic angina due to dynamic epicardial coronary obstruction. The hyperactivity of coronary vascular smooth muscle is often exacerbated by vasoconstrictor stimuli and environmental factors such as smoking, cold exposure, emotional stress or hyperventilation [64]. Diagnosis is characterized by a ≥90% reduction in epicardial artery diameter in response to a high dose of ACH with positive angina and ischemic changes on an ECG. Epicardial vasospastic angina can coexist with CMD and is typically associated with a worse prognosis [65].

## 6. Current Treatment of Coronary Microvascular Dysfunction

The lack of epicardial coronary disease often leads to misdiagnosis and inadequate treatment. Despite the absence of obstructive coronary artery disease, INOCA may have significant impacts on long-term prognosis, leading to increased hospitalizations, impaired quality of life, and increased morbidity and mortality [2,4,66]. A systematic review demonstrated a two- to four-fold increase in the risk of MACE in patients with endothelial-independent CMD, and a two-fold increase in risk in patients with endothelial-dependent CMD [67].

Guidelines for the treatment of INOCA and CMD are lacking due to the paucity of data [68]. Due to the concurrent occurrence of atherosclerosis and CMD, treatment should focus on the management of risk factors such as hypertension, diabetes, hyperlipidemia, and smoking cessation in order to reduce the progression of CMD [69]. Cardiac rehabilitation may also be beneficial in improving blood pressure, body mass index and exercise capacity [50]. However, treatments targeting the underlying pathology in CMD are limited, and data are limited to small cohorts. The WISE trial showed that angiotensin-converting enzyme inhibitors (ACEi) improve CFR and AF, and may reduce small vessel remodeling in CMD [70]. Furthermore, ACEi and angiotensinogen receptor blockers (ARB) can be combined with calcium channel blockers (CCBs) and beta blockers (BB) to provide better blood pressure control. Statins can also be beneficial in regard to their pleiotropic effects, including mitigating vascular inflammation in patients with reduced CFR [71]. Given the strong association with atherosclerosis, patients may benefit from antiplatelet therapy, such as aspirin, even in non-obstructive CAD [72]. The Women’s Ischemia Trial to Reduce Events in Non-Obstructive Coronary Artery Disease (WARRIOR) trial (NCT03417388) is a multicenter study currently evaluating whether treatment with aspirin, ACEi and statins in INOCA patients reduces MACE by 20% [73].

Anginal symptoms associated with INOCA are difficult to treat, as typical anti-anginal treatment is often ineffective. BB with CCBs are often the first line of antianginal therapy in INOCA. The mechanism of action is decreased myocardial oxygen consumption. In addition, CCBs dilate vascular smooth muscle and therefore are commonly used in addition to BB in CMD patients, or as the first line in patients with coronary micro- and macrovascular spam [74]. Interestingly, nitrates may result in symptom exacerbation due to a stealing effect [75]. 

Additional treatment modalities exist, but further research is needed to determine efficacy and safety in CMD. Ranolazine can be combined with other first-line therapies to improve myocardial relaxation and ventricular compliance via a reduction in intracellular calcium through decreased sodium channel activation, though recent small trials have shown no significant benefit in patients with CMD [76]. Nicorandil causes vasodilation through nitrate and potassium channel activation, but may lead to life-threatening bradycardia due to hyperkalemia [77]. Ivabradine can be beneficial in patients with anginal symptoms by decreasing heart rate without impairing left ventricular contractility [78]. Aminophylline, a nonselective adenosine-receptor antagonist, may be beneficial in shunting blood away from well-perfused areas in CMD by modulating excess microcirculatory dilation, resulting in improvements in symptoms and exercise capacity [79]. Second-line therapies include: trimetazidine, a fatty acid metabolism inhibitor; fasudil, a rho kinase inhibitor [80]; L-arginine supplementation [81]; cognitive behavioral therapy [82]; low-dose tricyclic antidepressants [83]. In patients with persistent angina despite medications, enhanced external counterpulsation (EECP) may be beneficial. EECP relies on the sequential inflation and deflation of lower extremity pneumatic compression devices synchronized with the cardiac cycle, which has been shown to improve hemodynamics and angina [50]. The diagnosis and current treatment of CMD is summarized in Figure 2.

## 7. CD34 Therapy as a Novel Treatment for Coronary Microvascular Dysfunction

Unfortunately, 25% of patients with INOCA have refractory angina despite medications. Cell therapy using autologous stem cells expressing CD34^+^ is a novel therapeutic option for these INOCA patients with CMD and refractory angina. The recent two-center ESCaPE-CMD trial (NCT03508609), sponsored by Caladrius Biosciences, evaluated the efficacy and safety of autologous CD34^+^ cell therapy in 20 INOCA patients with endothelial-independent CMD, defined as CFR ≤ 2.5, and refractory angina [21]. Patients received G-CSF for 4 days followed by leukapheresis, but, in contrast to the previous refractory angina trials, the CD34^+^ cells were delivered via an intracoronary route. The study showed that a single intracoronary infusion of autologous CD34^+^ cells significantly increased the CFR from 2.08 ± 0.32 at baseline to 2.68 ± 0.79 at 6 months after treatment (*p* = 0.0045). This correlated clinically to decreased AF (4.42 to 2.02, *p* = 0.0036), improved CCS class (3.2 to 2.05, *p* < 0.001), and improved Seattle Angina Questionnaire (SAQ) and 36-item short-form survey (SF-36) scores (Figure 3) [21]. As with the previous CD34^+^ clinical trials, there were no cell-related adverse events, demonstrating the excellent safety and tolerability of CD34^+^ infusions. This study extends the prior findings from the pooled analysis of trials in patients with obstructive CAD and refractory angina, to INOCA patients with CMD and persistent angina. This has led to the phase 2, randomized, double-blinded, placebo-controlled FREEDOM trial (NCT04614467), sponsored by Caladrius Biosciences. The trial began enrollment in October 2020 with an anticipated 105 patients randomized 2:1 to CD34^+^ cell therapy, to establish a potential therapeutic role for endothelial progenitor cells in INOCA patients with endothelial-independent CMD (Figure 4). A single infusion of autologous CD34^+^ cells will be administered intracoronary. Patients in the treatment and placebo arms will undergo G-CSF cell mobilization, apheresis, and the intracoronary infusion of autologous CD34^+^ cells vs. placebo. The primary endpoint is a change in CFR at 6 months.

Improvements in coronary microvascular function, as measured by CFR after CD34^+^ cell therapy, has led to the hypothesis that CD34^+^ cells repair the microvasculature in patients with CMD similarly to ischemic tissue. CD34^+^ cells promote vascular repair and enhance angiogenesis in the microvasculature, which restores the microcirculation and improves myocardial tissue perfusion, as evidenced by the improvement in CFR. Further, studies on the mechanisms of autologous CD34^+^ stem cells in CMD are needed in the future.

## 8. Conclusions

Endothelial-independent CMD, associated with a reduced CFR in response to intracoronary adenosine, increases the risk of MACE, MI, heart failure hospitalizations, and mortality in INOCA patients. However, no effective specific therapy to date exists for CMD. Cell therapy using autologous CD34^+^ stem cells is a promising new therapeutic option for INOCA patients with CMD. Through the induction of capillary growth, direct incorporation into damaged vasculature, and the upregulation of proliferative cytokines, CD34^+^ stem cell therapy can enhance angiogenesis and restore the microcirculation in acute and chronic ischemia. Three consecutive randomized and double-blinded trials have demonstrated the safety and efficacy of the reduction in CCS class 3–4 refractory angina from obstructive CAD via the delivery of intramyocardial CD34^+^ stem cells, and a pooled analysis of these studies showed significant improvements in AF and TET and a reduction of mortality. Feasibility and safety studies on the use of autologous CD34^+^ cell therapy in INOCA patients with endothelial-independent CMD and persistent angina showed significant improvement in CFR in 6 months, with no cell-related adverse events. The use of CD34^+^ cell therapy in CMD and persistent angina will be determined by the results of the ongoing phase 2 FREEDOM trial (NCT04614467). Given the high prevalence and the lack of specific treatment options in patient with INOCA and underlying CMD, additional research is needed to identify novel therapies.

## Figures and Tables

**Figure 1 cells-10-01137-f001:**
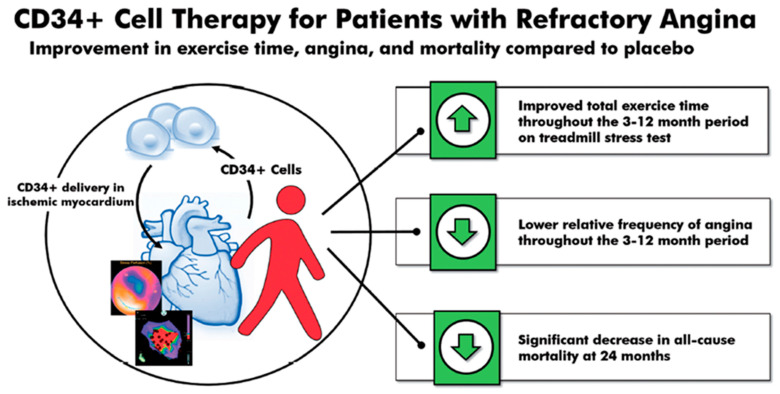
CD34^+^ cell therapy for patients with obstructive coronary artery disease and refractory angina. Results from the 2018 meta-analysis of three consecutive randomized, double-blinded, placebo-controlled trials in patients with obstructive coronary artery disease and Canadian Cardiovascular Society class 3–4 refractory angina showed that a single intracoronary infusion of autologous CD34^+^ cells significantly improved total exercise time, decreased angina frequency, and decreased all-cause mortality.

**Figure 2 cells-10-01137-f002:**
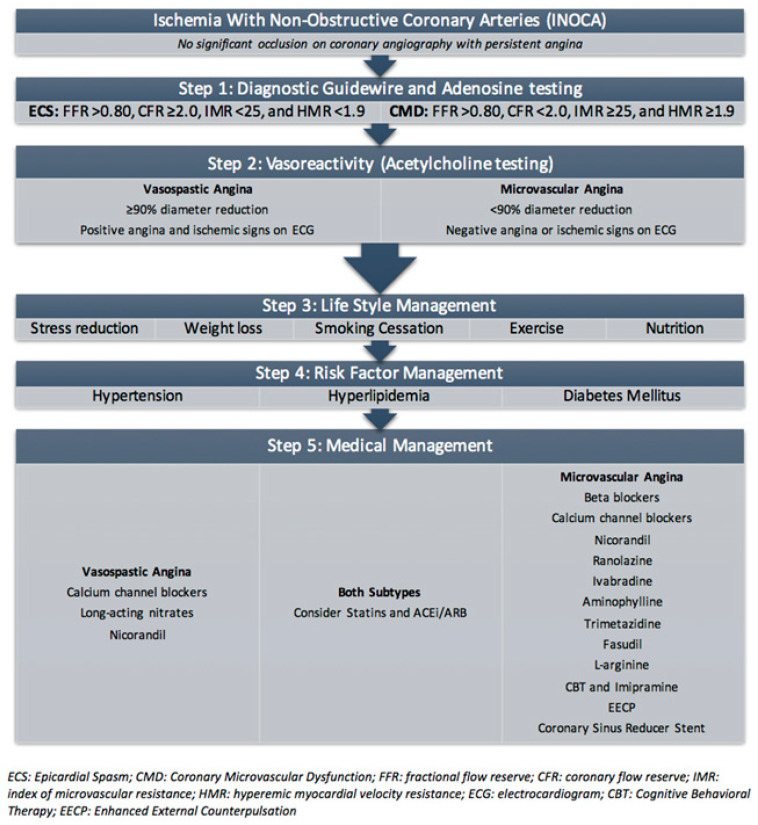
Diagnosis and treatment of coronary microvascular dysfunction.

**Figure 3 cells-10-01137-f003:**
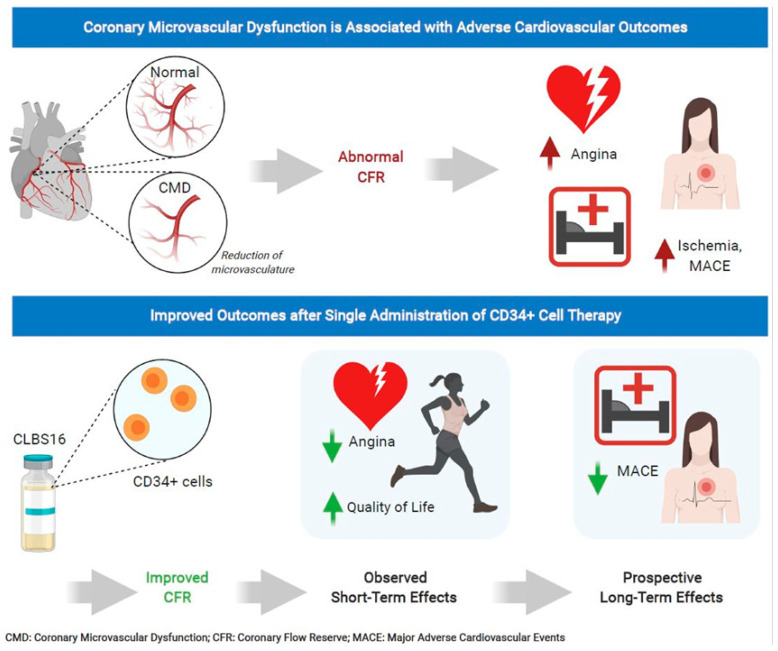
CD34^+^ cell therapy for patients with coronary microvascular dysfunction and refractory angina with no obstructive coronary artery disease. Results from the phase 1 ESCaPE-CMD trial (NCT03508609) showed that a single intracoronary infusion of autologous CD34^+^ cells in patients with coronary microvascular dysfunction and refractory angina with no obstructive coronary artery disease significantly improved coronary flow reserve, decreased angina frequency, and improved quality of life at 6 months.

**Figure 4 cells-10-01137-f004:**
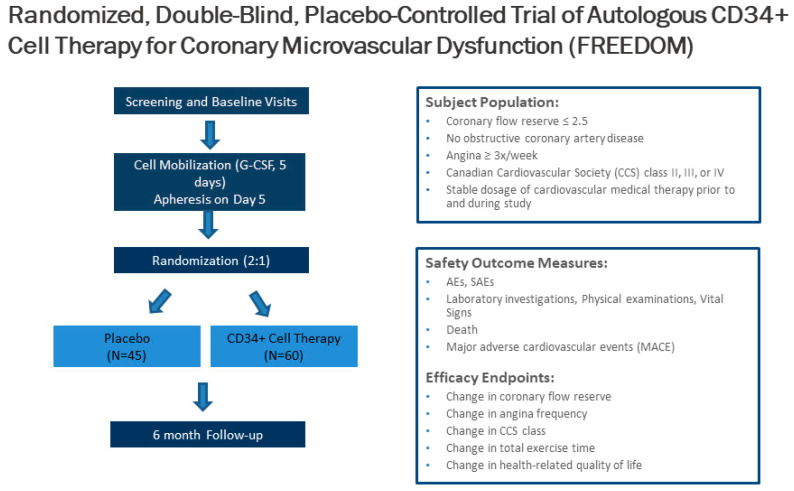
Trial design of the CD34^+^ cell therapy FREEDOM trial. The FREEDOM trial (NCT04614467) is a randomized, double-blinded, placebo-controlled trial of CD34^+^ cell therapy for patients with coronary microvascular dysfunction and refractory angina with no obstructive coronary artery disease.

## Data Availability

Not applicable.

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
