# Peer review of "Angiogenic CD34 Stem Cell Therapy in Coronary Microvascular Repair—A Systematic Review"

_cells, 2021, doi:10.3390/cells10051137_

Round 1

Reviewer 1 Report

This review deals with an important topic and reports interesting data that could be the basis for an efficient treatment in cases of Ischemia with non-obstructive coronary arteries. However, the manuscript needs much improvement and clarification.

In the first place, it would be better not to introduce too many abbreviations in the abstract.

Explain better the difference between endothelial independent and dependent CMD.

The authors could report the techniques used for the isolation of CD34 + from plasma.

Line 89: Interestingly, this was only shown in diabetic mice and not in the non-diabetic controls, possibly related to rapid healing time of the non-diabetic mice…..Explain better.

Are there any data in preclinical studies with CD34 + for the treatment of ischemic disease showing an increase in cell cycle speed with cell growth or formation of angiomas? According to the authors, there is no risk of uncontrolled cell growth of CD34 + inoculated into a tissue?

The paragraph "Clinical Studies on CD34 Therapy for Ischemic Disease" should be better organized. Treatments with C34 + in various types of patients are listed. Authors should discuss more the reported data, perhaps comparing them. In addition, subtitles should be included. The present form is dispersive and difficult to read.

As reported in the legend, fig. 1 shows how CD34 + cell treatment improves the quality of life of patients with obstructive coronary artery disease and refractory angina. In the text, however, it is not clear that these are patients with obstructive coronary artery disease.

Line 242 “Chronic hyperglycemia in diabetes can cause endothelium-dependent and endothelium–independent reduction in coronary vasodilatory capacity”. Diabetes exerts many other alterations on microcirculation. Clarify better.

Also the paragraph "Nonobstructive coronary artery disease and coronary microvascular dysfunction" should be divided into subtitles. For example, the part concerning the risk factors should be separated from the diagnosis .......Furthermore, the part of the diagnosis is too long and not adequate for the purpose of the study with the risk of moving away from the main topic.

It also seems inappropriate in this context to dwell too much on current treatments for CMD.

It would be enough to briefly report them and discuss in more depth the treatment with CD34 + cells in a single section.

Line 400: “A recent 2-center trial evaluated the efficacy and safety of autologous CD34 + cell therapy in INOCA patients with endothelial-independent CMD, defined as CFR≤2.5, and persistent angina. 83” . 83 does not appear in the Reference.

CD 34 Therapy as a Novel Treatment for Coronary Microvascular Dysfunction.” Who do the studies related to the paragraph belong to? Even in the legend of figures 3 and 4 the authors do not appear.

The authors always talk about Endothelial-independent CMD even if they initially mentioned also Endothelial-dependent CMD. Are there data on this last microvascular pathology as well? Was treatment with CD34 + cells also performed in this case? Authors should better discuss this aspect.

Author Response

This review deals with an important topic and reports interesting data that could be the basis for an efficient treatment in cases of Ischemia with non-obstructive coronary arteries. However, the manuscript needs much improvement and clarification.

Reply: We appreciate the Reviewer’s supportive comments and have substantially improved the manuscript as suggested.

  1. In the first place, it would be better not to introduce too many abbreviations in the abstract.

Reply: Thank you for this suggestion. We have made the suggested changes, keeping only two abbreviations to meet the abstract word limit.

  1. Explain better the difference between endothelial independent and dependent.

Reply: We agree this is an important point. Endothelium-dependent coronary microvascular dysfunction stems from an inability of endothelial cells to produce vasodilatory substrates, thus blunting adequate myocardial perfusion during stress. Conversely, endothelium-independent CMD results from an inability of smooth muscles to dilate despite adequate substrate availability. We have added this to the introduction.

  1. The authors could report the techniques used for the isolation of CD34 + from plasma.

Reply: Our focus is on the clinical aspects.  The techniques used for the isolation of CD34+ from plasma is proprietary information and is beyond the scope of our review.

  1. Line 89: Interestingly, this was only shown in diabetic mice and not in the non-diabetic controls, possibly related to rapid healing time of the non-diabetic mice…..Explain better.

Reply: The study found that diabetic mice with CD34+ injections had a significantly higher rate of flow restoration as compared to mice injected with CD34- cells following ligation and transection of the femoral artery. This was only shown in diabetic mice and not in the non-diabetic controls. Why exogenous CD34+ cells helped accelerate flow restoration in diabetic, but not nondiabetic, mice is unclear. Additional experimentation to determine a causal link was conducted but no link between insulin or glucose was established. The authors concluded that it is possible that exogenous cells are of little value to animals that already have a fully functioning complement of CD34+ cells and can help only in animals with compromised CD34+ cell function or with a smaller complement of angioblasts. Also, flow is restored so rapidly in the mouse model, it is possible that the measurements were not sensitive enough to pick up a CD34+ cell enhancement in nondiabetic mice. This possibility is supported by recent data indicating that cultured CD34+ cells can accelerate revascularization in an ischemic limb in nondiabetic mice (Kalka et al.).” We have made edits to reflect this.

  1. Are there any data in preclinical studies with CD34 + for the treatment of ischemic disease showing an increase in cell cycle speed with cell growth or formation of angiomas? According to the authors, there is no risk of uncontrolled cell growth of CD34 + inoculated into a tissue?

Reply: This is an important point regarding the safety profile of CD34+ treatment. Based on published data, CD34+ treatment has an excellent safety profile. No studies have reported an increased risk of uncontrolled cell growth of CD34+ inoculated tissues or increased formation of angiomas. We have added this point to the manuscript regarding the safety profile of CD34+ treatment for refractory angina.

  1. The paragraph "Clinical Studies on CD34 Therapy for Ischemic Disease" should be better organized. Treatments with C34 + in various types of patients are listed. Authors should discuss more the reported data, perhaps comparing them. In addition, subtitles should be included. The present form is dispersive and difficult to read.

Reply: We appreciate the reviewer comments and have added subtitles as the reviewer suggested to make the section on Clinical Studies on CD34 Therapy for Ischemic Disease more organized.

  1. As reported in the legend, fig. 1 shows how CD34 + cell treatment improves the quality of life of patients with obstructive coronary artery disease and refractory angina. In the text, however, it is not clear that these are patients with obstructive coronary artery disease.

Reply: Thank you for allowing us to clarify this important point. As the reviewer correctly points out the 2018 meta-analysis of these 3 randomized, double-blinded trials (n=304) – Phase I and II ACT-34, ACT-34 extension and Phase III RENEW – showed an improvement in TET, AF, and MACE in patients with obstructive coronary artery disease and refractory angina receiving intramyocardial autologous CD34+ cell therapy. We clarified the results pertain to patients with obstructive coronary artery disease and refractory angina.

  1. Line 242 “Chronic hyperglycemia in diabetes can cause endothelium-dependent and endothelium–independent reduction in coronary vasodilatory capacity”. Diabetes exerts many other alterations on microcirculation. Clarify better.

Reply: Chronic hyperglycemia in diabetes can cause endothelium-dependent and endothelium–independent reduction in coronary vasodilatory capacity potentially through hyperglycemia mediated formation of oxygen-derived free radicals which inactivated endothelium-derived nitric oxide. The sentence was modified to reflect this explanation.

  1. Also the paragraph "Nonobstructive coronary artery disease and coronary microvascular dysfunction" should be divided into subtitles. For example, the part concerning the risk factors should be separated from the diagnosis .......Furthermore, the part of the diagnosis is too long and not adequate for the purpose of the study with the risk of moving away from the main topic. It also seems inappropriate in this context to dwell too much on current treatments for CMD. It would be enough to briefly report them and discuss in more depth the treatment with CD34 + cells in a single section –

Reply: We appreciate the reviewer comments and have added subtitles to to make the INOCA and CMD sections more organized. This review focuses on CD34 stem cell therapy as a novel treatment for INOCA/CMD thus we felt that diagnosis and current treatment of CMD was important to highlight limitations of treatments which have been studied so far.  We have made edits to shorten the diagnosis and treatment sections as suggested by the reviewer.

  1. Line 400: “A recent 2-center trial evaluated the efficacy and safety of autologous CD34 + cell therapy in INOCA patients with endothelial-independent CMD, defined as CFR≤2.5, and persistent angina. 83” . 83 does not appear in the Reference.

Reply: Thank you for bringing this error to our attention. The manuscript is currenty under review. We substituted the ESCaPE-CMD phase 2 trial clinicaltrials.gov ID (NCT03508609) as a reference.

  1. CD 34 Therapy as a Novel Treatment for Coronary Microvascular Dysfunction.” Who do the studies related to the paragraph belong to? Even in the legend of figures 3 and 4 the authors do not appear.

Reply: Thank you for allowing us to clarify this point. The ESCaPE-CMD phase 2 trial (NCT03508609) and the FREEDOM trial (NCT04614467) were both sponsored by Caladrius Biosciences. The ESCAPE-CMD trial was presented as a late-breaking trial at the SCAI Meeting 2020 and is under review at Circulation: Cardiovascular Interventions.  We have added the Clinical Trials identifiers of the 2 trials since there are no publications at this time.

  1. The authors always talk about Endothelial-independent CMD even if they initially mentioned also Endothelial-dependent CMD. Are there data on this last microvascular pathology as well? Was treatment with CD34 + cells also performed in this case? Authors should better discuss this aspect.

Reply: Thank you for allowing us to clarify. In the review we discuss both endothelial-independent and endothelial-dependent CMD to give a complete overview of CMD. The intracoronary CD34 stem cell therapy trials used endothelium-independent CMD as the principle enrollment criteria (abnormal CFR ≤2.5). Typical antianginal medications are more effective in endothelium-depended CMD.

Reviewer 2 Report

The review is interesting and it provides a wide overview on studies about the use of Angiogenic CD34 Stem Cell Therapy in the coronary microvascular repair. Coronary microvascular dysfunction is an understimated pathophysiological mechanisms in the setting of ischemic heart disease as well as heart failure. This aspect is associated with the lack of a specific therapy againts CMD. 

Some aspects should be improved:

1) please discuss about the role of microvascular dysfunction in the ischemia reperfusion damage (microvascular hemorrage, no reflow etc etc) (see Int J Mol Sci 2020 Oct 30;21(21):8118. doi: 10.3390/ijms21218118 and Am J Physiol Heart Circ Physiol  2018 Sep 1;315(3):H550-H562. doi: 10.1152/ajpheart.00183.2018.)

2) please discuss the role of coronary ion channels and their involvement in ischemic heart disease. Because of their importance in the coronary blood flow regulation, they may represent a target in the therapy against ischemic heart disease ( see Eur J Prev Cardiol 2020 Jun 2;2047487320926780. doi: 10.1177/2047487320926780 and Int J Mol Sci 2020 Apr 30;21(9):3167.doi: 10.3390/ijms21093167)

3) please define all the abbreviations first time they appear in the text, abstract and figures.

4) there are some mistakes in the refrences order and bibliography (in the text there is ref number 83 that is unavailable in the reference list). Please check in the text that the reference numbers are in progressive order.

Author Response

  1. Please discuss about the role of microvascular dysfunction in the ischemia reperfusion damage (microvascular hemorrhage, no reflow etc) (see Int J Mol Sci 2020 Oct 30;21(21):8118. doi: 10.3390/ijms21218118 and Am J Physiol Heart Circ Physiol 2018 Sep 1;315(3):H550-H562. doi: 10.1152/ajpheart.00183.2018.)

Reply: Thank you for raising this important point about the role of microvascular dysfunction in the ischemia reperfusion damage. Reperfusion damage after percutaneous coronary stenting can cause microvascular injury which increases microvascular permeability and results in subsequent endothelium-dependent vasoconstriction and impaired vasodilation due to vasoactive substrate washout. We have added this to the section “Risk Factors in INOCA and CMD.”

  1. Please discuss the role of coronary ion channels and their involvement in ischemic heart disease. Because of their importance in the coronary blood flow regulation, they may represent a target in the therapy against ischemic heart disease (Eur J Prev Cardiol 2020 Jun 2;2047487320926780. doi: 10.1177/2047487320926780 and Int J Mol Sci 2020 Apr 30;21(9):3167.doi: 10.3390/ijms21093167)

Reply: As the reviewer notes, genetic variants in voltage-gated potassium channels, such as the SUR2 subunit, can also alter vascular tone via a disconnect between intracellular calcium release and changes in membrane excitability due to intracellular metabolism. We have added this to the session “Risk Factors in INOCA and CMD”

  1. Please define all the abbreviations first time they appear in the text, abstract and figures – re-look through abbreviations

Reply: As suggested, we have reviewed all abbreviations to ensure that they are defined.

  1. There are some mistakes in the references order and bibliography (in the text there is ref number 83 that is unavailable in the reference list). Please check in the text that the reference numbers are in progressive order. – RECHECK REFERENCES

Reply: We appreciate the reviewer bringing this to our attention. We have reviewed all the references and ensure that they are in the correct order. The Caladrius trial in reference 83 has not been published so we have removed the reference number and added the ESCaPE-CMD phase 2 trial clinicaltrials.gov ID (NCT03508609).

Round 2

Reviewer 1 Report

I disagree that the author's focus is purely clinical. Moreover there is no property whatsoever when publishing studies. However, the authors are not obliged to show methodological aspects.

In the text, reference numbers should be placed before the punctuation; for example [1],

In “CD 34 Therapy as a Novel Treatment for Coronary MicrovascularDysfunction.” and fig. 3, 4 the authors show data that are presented at the SCAI Meeting 2020, and that are under review at Circulation. The authors said that there are no publications at the moment. The authors should know that is not correct to show data without reporting them in the references, especially if these data are under review in another journal. Therefore, the authors have to choose, or insert a reference (conference proceedings, accepted studies) or have to delete the entire section.

Author Response

  1. I disagree that the author's focus is purely clinical. Moreover there is no property whatsoever when publishing studies. However, the authors are not obliged to show methodological aspects. 

Reply: We thank the reviewer for his comments, however for this “invited review” we were asked to focus on the clinical implications of CD34+ cells. Although we reviewed the preclinical data, our focus is not the methodological aspects of obtaining CD34 cells given that is not our expertise as clinical investigators. Instead we focused on the clinical trial data on the use of CD34+ cells that were obtained using different methods. Our goal was to review the requested topic, which is the novel use of CD34+ cells in coronary microvascular repair, a clinical syndrome currently with limited treatment options.

  1. In the text, reference numbers should be placed before the punctuation; for example [1], 

Reply: We made the changes as suggested.

  1. In “CD 34 Therapy as a Novel Treatment for Coronary MicrovascularDysfunction.” and fig. 3, 4 the authors show data that are presented at the SCAI Meeting 2020, and that are under review at Circulation. The authors said that there are no publications at the moment. The authors should know that is not correct to show data without reporting them in the references, especially if these data are under review in another journal. Therefore, the authors have to choose, or insert a reference (conference proceedings, accepted studies) or have to delete the entire section. 

Reply: Thank you for allowing us to clarify this important point. Figure 3 was designed to illustrate the concept that coronary microvascular dysfunction leads to abnormal CFR which is associated with increase in angina hospitalizations and MACE which was been shown in multiple studies as referenced in our review. It also summarizes the findings presented at Late Breaking trials at SCAI from the ESCaPE-CMD trial (NCT03508609) that are publicly available. Figure 4 is the trial design for the ongoing FREEDOM clinical trial (NCT04614467) to study CD34+ cells in patients with CMD and refractory angina and includes no data. We have included the clinical trial.gov numbers for both trials as is customary. As suggested by the reviewer we have included the link to the SCAI presentation. Depending on the timing of the publication the full published article may well be available. We were invited to write this review article to specifically address this novel approach and to describe the results of the initial trials and ongoing Phase 2 double blind randomized trial for CD34+ cell therapy in coronary microvascular dysfunction. Deleting this data would be detrimental to the review and its purpose.

Reviewer 2 Report

Authors correctly answered my comments. 

Author Response

Thank you